

# Increased chronic kidney disease development and progression in diabetic patients after appendectomy: a population-based cohort study

Chin-Hua Chang[1,*], Chew-Teng Kor[2,3,*], Chia-Lin Wu[1,4,5], Ping-Fang Chiu[1,6], Jhao-Rong Li[1], Chun-Chieh Tsai[1], Teng-Hsiang Chang[1] and Chia-Chu Chang[1,4,6,7,8]

[1] Department of Internal Medicine, Changhua Christian hospital, Changhua, Taiwan
[2] Department of Internal Medicine Research Center, Changhua Christian hospital, Changhua, Taiwan
[3] Graduate Institute of Statistics and Information Science, National Changhua University of Education, Changhua, Taiwan
[4] Department of Environmental and Precision Medicine Laboratory, Changhua Christian Hospital, Changhua, Lithuania
[5] Institute of Clinical Medicine, National Yang-Ming University, Taipei, Taiwan
[6] School of Medicine, Chung-Shan Medical University, Taichung, Taiwan
[7] Department of Internal Medicine, Kuang Tien General Hospital, Taichung, Taiwan
[8] Department of Nutrition, Hungkuang University, Taichung, Taiwan
[*] These authors contributed equally to this work.

## ABSTRACT

**Background**. The vermiform appendix serves as a "safe house" for maintaining normal gut bacteria and appendectomy may impair the intestinal microbiota. Appendectomy is expected to profoundly alter the immune system and modulate the pathogenic inflammatory immune responses of the gut. Recent studies have shown that a dysbiotic gut increases the risk of cardiovascular disease and chronic kidney disease (CKD). Therefore, we hypothesized that appendectomy would increase the risk of CKD.

**Methods**. This nationwide, population-based, propensity-score-matched cohort study included 10,383 patients who underwent appendectomy and 41,532 propensity-score-matched controls. Data were collected by the National Health Insurance Research Database of Taiwan from 2000 to 2013. We examined the associations between appendectomy and CKD and end-stage renal disease (ESRD). The major outcome was a new diagnosis of CKD based on an outpatient diagnosis made at least three times or hospital discharge diagnosis made once during the follow-up period. ESRD was defined as undergoing dialysis therapy for at least 90 days, as in previous studies.

**Results**. The incidence rates of CKD and ESRD were higher in the appendectomy group than in the control cohort (CKD: 6.52 vs. 5.93 per 1,000 person-years, respectively; ESRD: 0.49 vs. 0.31 per 1,000 person-years, respectively). Appendectomy patients also had a higher risk of developing CKD (adjusted hazard ratio [aHR] 1.13; 95% CI [1.01–1.26]; $P = 0.037$) and ESRD (aHR 1.59; 95% CI [1.06–2.37]; $P = 0.024$) than control group patients. Subgroup analysis showed that appendectomy patients with concomitant diabetes mellitus (aHR 2.08; $P = 0.004$) were at higher risk of incident ESRD than those without diabetes mellitus. The interaction effects of appendectomy

Corresponding author
Chia-Chu Chang,
chiachu@cch.org.tw,
27509@cch.org.tw

and diabetes mellitus were significant for ESRD risk ($P = 0.022$); no interaction effect was found for CKD risk ($P = 0.555$).

**Conclusions**. Appendectomy increases the risk of developing CKD and ESRD, especially in diabetic patients. Physicians should pay close attention to renal function prognosis in appendectomy patients.

# INTRODUCTION

Chronic kidney disease (CKD) is a global public health problem affecting up to 10% of the population worldwide (*Nallu et al., 2017*). CKD influences kidney structure and function and typically results in end-stage renal disease (ESRD). Better approaches for the prevention, early detection, and treatment of CKD are needed (*Gansevoort et al., 2013*). Therefore, it is important to recognize the risk factors for CKD.

The vermiform appendix serves as a "safe house" for maintaining normal gut bacteria and can provide support for bacterial growth (*Bollinger et al., 2003*; *Bollinger et al., 2007*). *Guinane et al. (2013)* found that the appendix possesses a microbial diversity sufficient to reconstitute the microbiome of the colon. The appendix also contains the highest concentration of gut-associated lymphoid tissue, which has numerous functions (*Sanders et al., 2013*). Appendectomy is expected to profoundly alter the immune system and modulate the pathogenic inflammatory immune responses of the gut (*Sanders et al., 2013*). Furthermore, studies have shown that appendectomy-related impairment of the microbiota may lead to dysbiosis and induce various diseases, including ulcerative colitis, Crohn's disease, *Clostridium difficile* infection, colorectal cancer, rheumatoid arthritis, and cardiovascular disease (*Sanders et al., 2013*; *Tzeng et al., 2015*; *Roblin et al., 2012*; *Wu et al., 2015*).

To date, there is no evidence confirming that appendectomy contributes to CKD. In this study, we evaluated the correlation between prior appendectomy and the occurrence of CKD and ESRD.

# MATERIALS AND METHODS

## Data source

Data for this study were retrieved from the Taiwan National Health Insurance Research Database (NHIRD), which contains data from all medical claims in the Taiwan National Health Insurance Program. This insurance program started in 1995 and covers >99% of the Taiwanese population of approximately 23 million people. Our research dataset was the Longitudinal Health Insurance Database 2005 (LHID2005), which includes the data from one million patients randomly selected from the NHIRD in 2005 and longitudinally linked with the NHIRD from 1996 to 2013. LHID2005 is managed and established by the National
Health Insurance. Diagnostic codes used in the LHID2005 to identify diseases are taken from the International Classification of Diseases, Ninth Revision, Clinical Modification (ICD-9-CM). These diagnostic codes have been shown to have high accuracy and validity. (*Cheng et al., 2011*; *Cheng et al., 2014*; *Hsu et al., 2014*). This study was approved after a full ethical review by the Institutional Review Board of the Changhua Christian Hospital (approval number 151219), which waived the need for consent. Data were accessed anonymously.

## Study population

We used a four-year look-back period (1996–1999) for newly identified appendectomy patients. From the hospitalization database of the LHID2005, we identified patients aged 18 to 100 years who underwent appendectomy from 2000 to 2013 (ICD-9-CM codes 47.0 and 47.1). The index date was defined as the date of appendectomy. Each identified appendectomy patient was randomly matched with four control patients according to propensity score. We excluded patients with a history of CKD before the index date, those aged <18 or >100 years, those with incomplete demographic data, and those with fewer than 30 days of follow-up.

## Outcomes and relevant variables

Outcomes and comorbidities were diagnosed according to ICD-9 codes. The major outcome was a new diagnosis of CKD (ICD-9 codes 580–585), based on an outpatient diagnosis made at least three times or hospital discharge diagnosis made once during the follow-up period. ESRD was defined as undergoing dialysis therapy for at least 90 days, as in previous studies (*Wu et al., 2016*; *Wu et al., 2017*; *Weng et al., 2017*).

The following comorbidities potentially related to CKD were investigated: hypertension (ICD-9 401–405), diabetes mellitus (DM; ICD-9 250), hyperlipidemia (ICD-9 272), coronary artery disease (CAD; ICD-9 410–414), congestive heart failure (CHF; ICD-9 402–404, 425.4–425.9, 428), arrhythmia (ICD-9 427), stroke (ICD-9 430–438), peripheral artery occlusive disease (PAOD; ICD-9 443–444), gout (ICD-9 274), and chronic obstructive pulmonary disease (COPD; ICD-9 416.8, 416.9, 490–505, 506.4, 508, 508.1). Charlson Comorbidity Index (CCI) scores were used to measure the severity of comorbidities at baseline (*Deyo, Cherkin & Ciol, 1992*). Some long-term medications have been associated with renal outcomes, including antidiabetic agents, angiotensin-converting-enzyme inhibitors, angiotensin II receptor blockers, beta blockers, diuretics, nonsteroidal anti-inflammatory drugs (NSAIDs), analgesic drugs other than NSAIDs, and statins. All study subjects were followed from the index date until the first diagnosis of CKD or ESRD, withdrawal from the insurance system, or at the end of 2013.

## Statistical analysis

Demographic and clinical characteristics in the appendectomy and non-appendectomy cohorts are presented as number (percentage) and mean ± standard deviation (SD). Differences in categorical and continuous variables were compared between the cohorts with chi-square tests and Student's $t$-tests, respectively. To reduce the potential bias from uncontrolled confounding, we performed propensity-score-matching studies. To achieve

a maximum power to illustrate a significant association between appendectomy and CKD, we used a 1:4 propensity-score-matching approach in our study. Propensity scores were calculated with multivariate logistic regression to predict the likelihood of undergoing appendectomy (Table S1). The incidence of CKD was calculated as the number of CKD events occurring during the follow-up year, divided by the total follow-up person-years for all subjects.

The Cox proportional hazards regression model with competing risks of death was applied to estimate the hazard ratio (HR) and 95% confidence interval (CI) for risk of incident CKD in patients after appendectomy compared with the non-appendectomy cohort. Multivariate Cox's hazards analysis was used with the Fine–Grey competing risks model to estimate adjusted hazard ratios (aHRs), adjusting for the following confounders: appendectomy, demographic factors (age, sex, clinical visit frequency, and income), comorbidities (hypertension, DM, hyperlipidemia, gout, stroke, CAD, CHF, arrhythmia, PAOD, COPD, and CCI score), and long-term use of medications (statins, antidiabetic drugs, antihypertensive drugs, NSAIDs, and analgesic drugs other than NSAIDs). Subgroup analyses were used to distinguish CKD and ESRD risks in patients after appendectomy and in the control cohort according to various subpopulations. The cumulative incidence of CKD and ESRD in both study cohorts were examined with the Fine–Grey sub-distribution hazard approach and compared with Grey's tests. All statistical analyses were performed with SAS 9.4 software (SAS Institute Inc., Cary, NC, USA). A two-tailed $P$ value $<0.05$ was considered statistically significant.

# RESULTS

## Characteristics of the study population

Figure 1 shows a flowchart of the subject selection process and Table 1 shows the characteristics of the study population. This study comprised 51,915 patients, including 10,383 with new appendectomy and 41,532 propensity-score-matched controls not diagnosed with appendectomy. The follow-up times (mean $\pm$ SD) in these cohorts were $5.81 \pm 3.98$ years and $6.66 \pm 4.02$ years, respectively. There was no significant difference in sex or age between the appendectomy and non-appendectomy cohorts. The prevalence of comorbidities, including hypertension, DM, hyperlipidemia, CAD, CHF, arrhythmia, stroke, PAOD, gout, and COPD, were similar in both cohorts (Table 1). Compared with the control cohort, patients with appendectomy were slightly less likely to be taking long-term angiotensin-converting-enzyme inhibitors or angiotensin II receptor blockers (5.94% vs. 6.48%, $P = 0.0433$).

## Long-term risk of incident CKD and ESRD

During the follow-up period, the proportion of patients who developed CKD or ESRD was significantly higher among appendectomy patients than in the control cohort (CKD: 4.34% vs. 3.44%, $P < 0.0001$; ESRD: 0.34% vs. 0.18%, $P = 0.0019$; Table 1). The cumulative incidence curve of CKD and ESRD according to the Fine–Grey method was also significantly higher for subjects with appendectomy than for the control cohort (Grey's test: $P = 0.012$ for CKD, $P = 0.014$ for ESRD; Figs. 2A and 2B). The incidence rates of CKD and ESRD

Chang et al. (2018), *PeerJ*, DOI 10.7717/peerj.5019

**Table 1  Patient demographics.**

| | All patients (N = 707,842) | | | After match 1:4 (N = 51,915) | | |
|---|---|---|---|---|---|---|
| | Non-appendectomy (N = 697,459) | Appendectomy (N = 10,383) | *p*-value | Controls (N = 41,532) | Appendectomy (N = 10,383) | *p*-value |
| **Demographics** | | | | | | |
| Gender, male | 334,344 (47.94%) | 5,253 (50.59%) | <0.0001 | 21,249 (51.16%) | 5,253 (50.59%) | 0.2982 |
| Age | 43.53 ± 16.67 | 41.71 ± 16.38 | <0.0001 | 41.48 ± 16.39 | 41.71 ± 16.38 | 0.2044 |
| Monthly income (NTD) | 18,436.6 ± 15,974.1 | 17,975.8 ± 15,225.0 | 0.0022 | 17,886.9 ± 15,479.9 | 17,975.8 ± 15,225.0 | 0.5957 |
| Clinic visit frequency | 16.01 ± 14.30 | 17.04 ± 14.26 | <0.0001 | 17.02 ± 15.45 | 17.04 ± 14.26 | 0.8784 |
| **Comorbidities disease at baseline** | | | | | | |
| Hypertension | 114,369 (16.40%) | 1,551 (14.94%) | <0.0001 | 6,124 (14.75%) | 1,551 (14.94%) | 0.6209 |
| Diabetes mellitus | 47,722 (6.84%) | 647 (6.23%) | 0.0143 | 2,602 (6.27%) | 647 (6.23%) | 0.8991 |
| Hyperlipidemia | 69,706 (9.99%) | 923 (8.89%) | 0.0002 | 3,710 (8.93%) | 923 (8.89%) | 0.8898 |
| CAD | 44,464 (6.38%) | 642 (6.18%) | 0.4267 | 2,540 (6.12%) | 642 (6.18%) | 0.7978 |
| CHF | 9,419 (1.35%) | 134 (1.29%) | 0.5995 | 9,419 (1.35%) | 134 (1.29%) | 0.5995 |
| Arrhythmia | 25,794 (3.70%) | 425 (4.09%) | 0.0344 | 1,685 (4.06%) | 425 (4.09%) | 0.8676 |
| Stroke | 28,713 (4.12%) | 368 (3.54%) | 0.0035 | 1,458 (3.51%) | 368 (3.54%) | 0.8675 |
| PAOD | 5,265 (0.75%) | 64 (0.62%) | 0.1051 | 257 (0.62%) | 64 (0.62%) | 0.9777 |
| Gout | 36,749 (5.27%) | 525 (5.06%) | 0.3356 | 2,074 (4.99%) | 525 (5.06%) | 0.7936 |
| COPD | 70,848 (10.16%) | 1,046 (10.07%) | 0.7789 | 4,105 (9.88%) | 1,046 (10.07%) | 0.5620 |
| CCI score | 0.65 ± 1.33 | 0.88 ± 1.76 | <0.0001 | 0.86 ± 1.80 | 0.88 ± 1.76 | 0.2601 |
| **Long term medication use** | | | | | | |
| Anti-diabetic agents | 32,681 (4.69%) | 411 (3.96%) | 0.0005 | 1,595 (3.84%) | 411 (3.96%) | 0.5769 |
| Antihypertensive drug | 110,224 (15.80%) | 1,492 (14.37%) | <0.0001 | 5,924 (14.26%) | 1,492 (14.37%) | 0.7826 |
| Diuretics | 27,619 (3.96%) | 368 (3.54%) | 0.0310 | 1,557 (3.75%) | 368 (3.54%) | 0.3236 |
| ACEIs/ARBs | 51,323 (7.36%) | 617 (5.94%) | <0.0001 | 2,693 (6.48%) | 617 (5.94%) | 0.0433 |
| Beta-blockers | 54,829 (7.86%) | 724 (6.97%) | 0.0008 | 2,940 (7.08%) | 724 (6.97%) | 0.7062 |
| NSAIDs | 41,546 (5.96%) | 667 (6.42%) | 0.0460 | 2,593 (6.24%) | 667 (6.42%) | 0.4975 |
| Analgesic drugs other than NSAIDs | 23,206 (3.33%) | 441 (4.25%) | <0.0001 | 1,697 (4.09%) | 441 (4.25%) | 0.4594 |
| Statins | 24,292 (3.48%) | 279 (2.69%) | <0.0001 | 1,125 (2.71%) | 279 (2.69%) | 0.9031 |
| Propensity score | 0.015 ± 0.005 | 0.016 ± 0.008 | <0.0001 | 0.016 ± 0.008 | 0.016 ± 0.008 | 0.9999 |
| **Outcome** | | | | | | |
| CKD | 26,298 (3.77%) | 451 (4.34%) | 0.0024 | 1,430 (3.44%) | 451 (4.34%) | <0.0001 |
| ESRD | 1,544 (0.22%) | 35 (0.34%) | 0.0131 | 75 (0.18%) | 35 (0.34%) | 0.0019 |
| Death | 86,965 (12.47%) | 1,129 (10.87%) | <0.0001 | 5,300 (12.76%) | 1,129 (10.87%) | <0.0001 |
| Subsequent death after ESRD or CKD | 9,640 (1.38%) | 145 (1.40%) | 0.9346 | 533 (1.28%) | 145 (1.39%) | 0.3897 |
| Death without CKD or ESRD | 77,325 (11.09%) | 984 (9.48%) | <0.0001 | 4767 (11.48%) | 984 (9.47%) | <0.0001 |

**Notes.**

Abbreviations: NTD, new Taiwan dollars; ACEI, Angiotensin-converting-enzyme inhibitor; ARB, Angiotensin II receptor blocker; CAD, coronary artery disease; CHF, congestive heart failure; CCI, Charlson's comorbidity index; CKD, chronic kidney disease; ESRD, end-stage renal disease; NSAIDs, Non-steroidal anti-inflammatory drugs; PAOD, peripheral artery occlusive disease; COPD, chronic obstructive pulmonary disease.

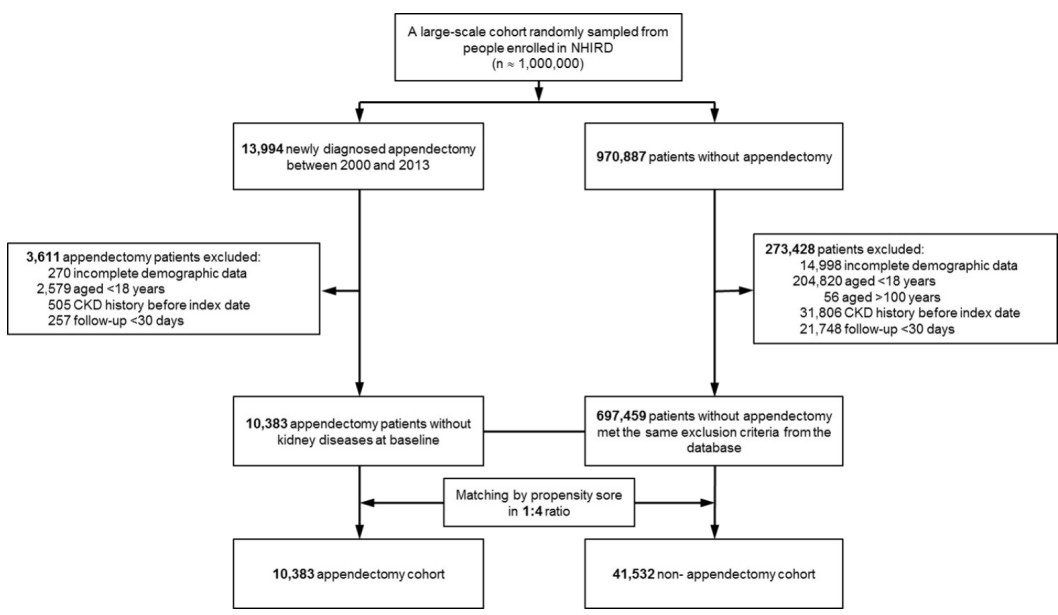

**Figure 1 Flowchart.** Flowchart of subject selection process.

were 6.52 and 0.49, respectively, per 1,000 person-years for appendectomy patients and 5.93 and 0.31, respectively, per 1000 person-years for the control cohort (Table 2). To evaluate the reliability of our results, we used three models to adjust the risk for incident CKD and ESRD among appendectomy patients compared with the control cohort (Table 2). According to propensity-score-matched data, the risks of CKD and ESRD were significantly higher among appendectomy patients than among controls (CKD: crude HR [cHR] 1.15, 95% CI [1.03–1.27], $P = 0.012$; ESRD: cHR 1.65, 95% CI [1.10–2.47], $P = 0.014$). Even after adjustment for the propensity score, the risk of CKD and ESRD remained higher in the appendectomy group than in the control cohort (CKD: adjusted HR (aHR) 1.13, 95% CI [1.01–1.25], $P = 0.029$; ESRD: aHR 1.62, 95% CI [1.09–2.42], $P = 0.018$). Finally, we performed multivariate analysis adjusting for all confounding variables listed in Table 1. The results of this analysis showed that the risks of CKD and ESRD were similarly higher in the appendectomy group than in the control cohort (CKD: aHR 1.13, 95% CI [1.01–1.26], $P = 0.037$; ESRD: aHR 1.59, 95% CI [1.06–2.37], $P = 0.024$).

## Subgroup analysis of CKD risk according to age, sex, comorbid conditions, and DM in appendectomy patients versus control cohort

Table 3 shows that the HR for incident CKD was significantly higher in the appendectomy group than in the control cohort only for males (aHR 1.16, 95% CI [1.00–1.34], $P = 0.045$) and for those without comorbid conditions (aHR 1.29, 95% CI [1.08–1.55], $P = 0.006$). The association between appendectomy and CKD risk was absent in female patients, in individual age groups, and in those with comorbid conditions (1–2 versus ≥ 3 comorbid diseases), including DM. The interactions between all stratified subgroups and appendectomy were not significant (all $P_{\text{interaction}} > 0.05$).

Chang et al. (2018), *PeerJ*, DOI 10.7717/peerj.5019

**Table 2  Incidence rate and risk of CKD and ESRD.** Incidence rate and risk of Chronic Kidney Disease (CKD) and End-Stage Renal Disease (ESRD) in patients with appendectomy and matched participants.

| | Events (no.) | PY[a] | Incidence[b] | Model 1 | | Model 2 | | Model 3 | |
|---|---|---|---|---|---|---|---|---|---|
| | | | | HR (95% CI) | *P*-value | Adj. HR (95% CI) | *P*-value | Adj. HR (95% CI) | *P*-value |
| **CKD** | | | | | | | | | |
| Control cohort | 1,430 | 241,152.6 | 5.93 (5.62, 6.24) | Reference | | Reference | | Reference | |
| Appendectomy cohort | 451 | 69,120.0 | 6.52 (5.92, 7.13) | 1.145 (1.030, 1.273) | 0.0120 | 1.125 (1.012, 1.251) | 0.0291 | 1.125 (1.007, 1.256) | 0.0365 |
| **ESRD** | | | | | | | | | |
| Control cohort | 75 | 245,459.0 | 0.31 (0.24, 0.37) | Reference | | Reference | | Reference | |
| Appendectomy cohort | 35 | 70,716.2 | 0.49 (0.33, 0.66) | 1.653 (1.107, 2.468) | 0.0140 | 1.622 (1.087, 2.421) | 0.0179 | 1.588 (1.064, 2.371) | 0.0237 |

**Notes.**

Model 1: Propensity-score matched.

Model 2: Adjusted for propensity score.

Model 3: Adjusted for all variables listed in Table 1.

[a]PY: person-years.

[b]per 1000 person-years.

HR, Hazard ratio; Adj. HR, adjusted hazard ratio; CI, confidence interval.

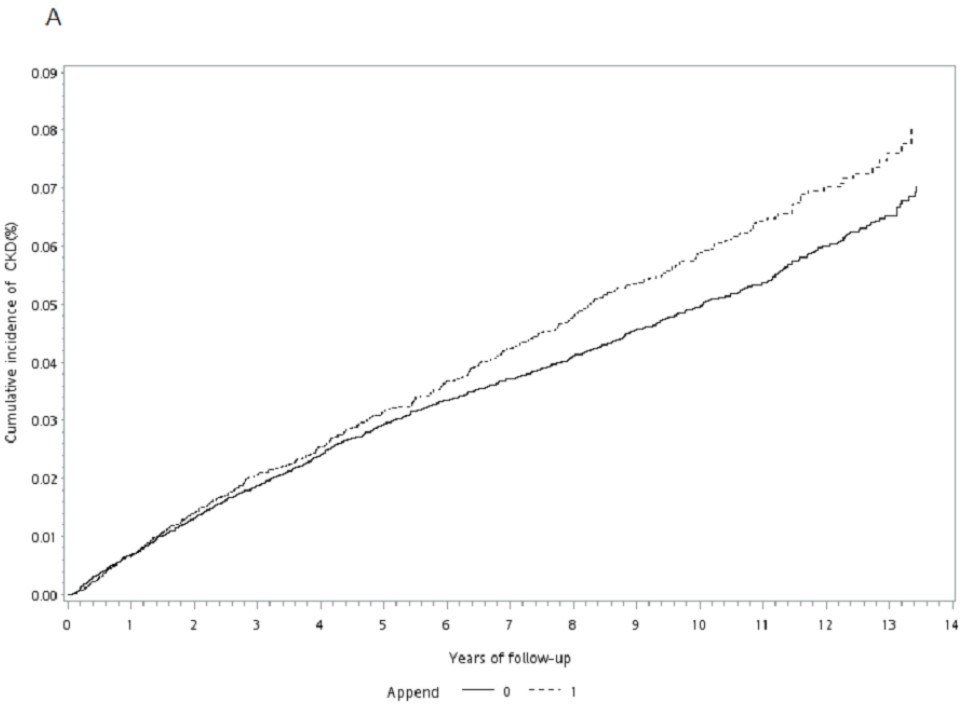

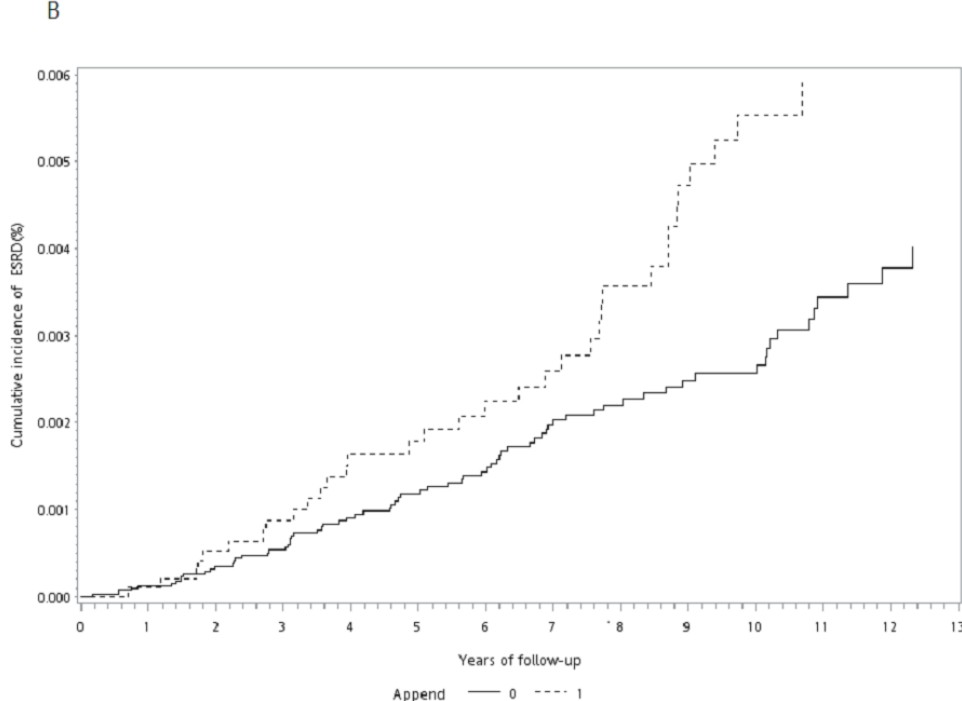

**Figure 2 Cumulative incidence curve of CKD and ESRD.** (A) Cumulative incidence curve of CKD for patients after appendectomy compared with control cohort. (B) Cumulative incidence curves of ESRD for patients after appendectomy compared with control cohort.

**Table 3 Subgroup analyses of chronic kidney disease risk.** Subgroup analyses of chronic kidney disease risk in patients with appendectomy and matched control group.

| Subgroup | Subjects without appendectomy | | Subjects with appendectomy | | Compared with control group | | | | |
|---|---|---|---|---|---|---|---|---|---|
| | Total no. | Event (no.) | Total no. | Event (no.) | aHR (95% CI)[a] | P-value | aHR (95% CI)[b] | P-value | $P_{interaction}$ |
| Sex | | | | | | | | | 0.4765 |
| Female | 20,283 | 592 | 5,130 | 193 | 1.080 (0.916, 1.274) | 0.3576 | 1.068 (0.901, 1.266) | 0.4467 | |
| Male | 21,249 | 838 | 5,253 | 258 | 1.172 (1.019, 1.347) | 0.0259 | 1.160 (1.003, 1.342) | 0.0452 | |
| Age, years | | | | | | | | | 0.8848 |
| <30 | 12,470 | 69 | 3,057 | 17 | 0.890 (0.521, 1.521) | 0.6696 | 0.933 (0.544, 1.600) | 0.8013 | |
| 30–65 | 24,693 | 780 | 6,176 | 248 | 1.133 (0.983, 1.306) | 0.0855 | 1.122 (0.970, 1.298) | 0.1225 | |
| ≥65 | 4,369 | 581 | 1,150 | 186 | 1.111 (0.942, 1.310) | 0.2110 | 1.096 (0.925, 1.298) | 0.2918 | |
| Comorbidities (no.) | | | | | | | | | 0.2755 |
| 0 | 29,761 | 446 | 7,314 | 160 | 1.265 (1.054, 1.518) | 0.0114 | 1.293 (1.078, 1.551) | 0.0057 | |
| 1–2 | 8,119 | 527 | 2,178 | 157 | 0.965 (0.808, 1.154) | 0.6990 | 0.953 (0.796, 1.142) | 0.6032 | |
| ≥3 | 3,652 | 457 | 891 | 134 | 1.084 (0.894, 1.313) | 0.4122 | 1.088 (0.895, 1.322) | 0.3968 | |
| Diabetes mellitus | | | | | | | | | 0.5547 |
| No | 38,930 | 1,020 | 9,736 | 313 | 1.091 (0.961, 1.238) | 0.1790 | 1.102 (0.968, 1.253) | 0.1420 | |
| Yes | 2,602 | 410 | 647 | 138 | 1.093 (0.900, 1.326) | 0.3698 | 1.122 (0.921, 1.367) | 0.2534 | |

**Notes.**
[a] Model was adjusted for propensity score.
[b] Model was adjusted for all variables listed in Table 1.

## Subgroup analysis of ESRD risk according to age, sex, comorbid conditions, and DM in appendectomy patients versus control cohort

Table 4 shows that the HR for incident ESRD was significantly higher for appendectomy patients than for the control cohort only among patients who were middle-aged (30 −64 years) (aHR 2.05, 95% CI [1.17–3.57], $P = 0.012$) and that there was a marginally higher risk among females (aHR 1.68, 95% CI [0.97–2.92], $P = 0.067$). The association between appendectomy and ESRD risk was absent for male patients, for other age groups (<30 years and ≥ 65 years), and for those with comorbid conditions (0, 1–2, and ≥ 3 comorbid diseases). However, the interactions between sex, age group, comorbid conditions, and appendectomy were not significant (all $P_{interaction} > 0.05$). In particular, appendectomy patients with concomitant DM were found to have a significantly increased risk of ESRD compared with the control cohort patients with DM (aHR 2.08, 95% CI [1.27–3.43], $P = 0.004$; Table 4). The interactions between appendectomy and DM for risk of ESRD were significant ($P = 0.022$).

## Interaction effects between appendectomy and comorbidities or medication use on CKD and ESRD risk

Interaction analyses for CKD risk are shown in Fig. 3A. Patients with appendectomy had a higher risk of CKD compared with the control cohort in most subgroups, except patients with stroke, gout, COPD, use of analgesic drugs other than NSAIDs, and use of statins. Interaction analyses for ESRD risk are shown in Fig. 3B. An increased risk of ESRD was also observed consistently in most subgroups, except those with stroke, use of NSAIDs,

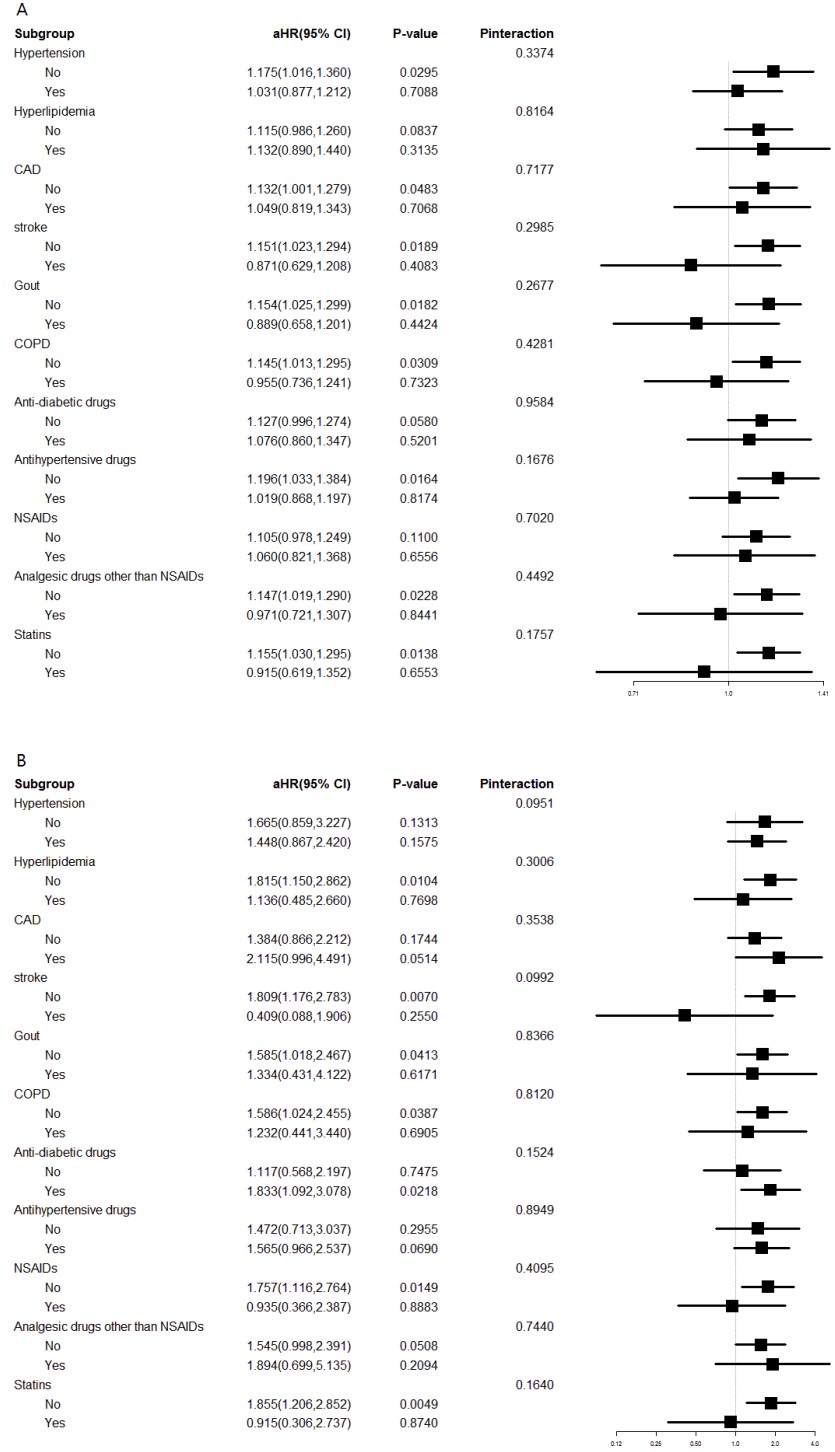

**Figure 3  Interaction effects between appendectomy and comorbidities or medication.** (A) Interaction effects between appendectomy and comorbidities or medication use on CKD risk. (B) Interaction effect between appendectomy and comorbidities or medication use on ESRD risk.

**Table 4  Subgroup analyses of end-stage renal disease risk.** Subgroup analyses of end-stage renal disease risk in patients with appendectomy and matched control group.

| Subgroup | Subjects without appendectomy | | Subjects with appendectomy | | Compared with control group | | | | |
|---|---|---|---|---|---|---|---|---|---|
| | Total | Event | Total | Event | aHR (95% CI)[a] | P-value | aHR (95% CI)[b] | P-value | $P_{interaction}$ |
| Sex | | | | | | | | | 0.9338 |
| Female | 20,283 | 40 | 5,130 | 20 | 1.546 (0.903, 2.648) | 0.1125 | 1.679 (0.965, 2.922) | 0.0666 | |
| Male | 21,249 | 35 | 5,253 | 15 | 1.624 (0.883, 2.988) | 0.1190 | 1.625 (0.863, 3.060) | 0.1325 | |
| Age, years | | | | | | | | | 0.0715 |
| <30 | 12,470 | 2 | 3,057 | 2 | 3.049 (0.389, 23.920) | 0.2887 | 3.792 (0.594, 24.204) | 0.1587 | |
| 30–65 | 24,693 | 34 | 6,176 | 21 | 2.118 (1.234, 3.635) | 0.0065 | 2.048 (1.173, 3.573) | 0.0117 | |
| ≥65 | 4,369 | 39 | 1,150 | 12 | 1.000 (0.526, 1.901) | 0.9994 | 0.937 (0.487, 1.800) | 0.8441 | |
| Comorbidities (no.) | | | | | | | | | 0.8698 |
| 0 | 29,761 | 13 | 7,314 | 6 | 1.583 (0.594, 4.216) | 0.3581 | 1.501 (0.559, 4.032) | 0.4204 | |
| 1–2 | 8,119 | 21 | 2,178 | 10 | 1.333 (0.636, 2.791) | 0.4463 | 1.300 (0.603, 2.801) | 0.5029 | |
| ≥3 | 3,652 | 41 | 891 | 19 | 1.618 (0.935, 2.797) | 0.0852 | 1.651 (0.959, 2.840) | 0.0702 | |
| Diabetes mellitus | | | | | | | | | **0.0222** |
| No | 38,930 | 35 | 9,736 | 8 | 0.804 (0.372, 1.735) | 0.5777 | 0.829 (0.386, 1.780) | 0.6302 | |
| Yes | 2,602 | 40 | 647 | 27 | 1.964 (1.207, 3.195) | 0.0066 | 2.082 (1.266, 3.425) | 0.0039 | |

**Notes.**
[a]Model was adjusted for propensity score.
[b]Model was adjusted for all variables listed in Table 1.

and use of statins. However, the interactions between appendectomy and concomitant chronic diseases or medication use were not significant for any of these subgroups (all $P_{interaction} > 0.05$; Figs. 3A and 3B).

## DISCUSSION

Acute appendicitis is one of the most common surgical emergencies, with a rate of approximately 10 cases per 10,000 people per year (*Buckius et al., 2012*). The microbiota in the large bowel can change after appendectomy, leading to dysbiosis (*Wu et al., 2015*). Gut dysbiosis and inflammation are underlying and linking factors between CKD and DM (*Sabatino et al., 2017*). Appendectomy could worsen pre-existing dysbiosis in diabetic patients, increasing the risk of ESRD development. In addition, appendectomy has been identified as a risk factor for the development of acute kidney injury (*Kim, Brady & Li, 2014*). Furthermore, many studies have suggested that appendectomy may increase the risk of ulcerative colitis, colon cancer, *C. difficile* colitis, and rheumatic arthritis (*Sanders et al., 2013*; *Tzeng et al., 2015*; *Roblin et al., 2012*; *Wu et al., 2015*). In this large, retrospective, population-based cohort study, we revealed that the risks of CKD (aHR 1.13, 95% CI [1.01–1.26], $P = 0.037$) and ESRD (aHR 1.59, 95% CI [1.06–2.37], $P = 0.024$) were higher among appendectomy patients than in the control cohort. These results support our supposition that there are associations between appendectomy and CKD and ESRD.

Recently, many studies have suggested that intestinal epithelial barrier structure and function are impaired in CKD patients (*Vaziri, Zhao & Pahl, 2016*; *Vaziri et al., 2013*). Furthermore, progressive loss of kidney function significantly contributes to intestinal

dysbiosis in CKD patients (*Jiang et al., 2017*; *Anders, Andersen & Stecher, 2013*). The presence of intestinal dysbiosis is associated with elevated concentrations of uremic toxins. (*Evenepoel et al., 2009*; *Pan & Kang, 2018*; *Ramezani & Raj, 2014*; *Koppe, Mafra & Fouque, 2015*). Several microbiota-derived uremic retention solutes have been identified, including p-cresyl sulfate and indoxyl sulfate (*Poesen et al., 2016*). In experimental studies, indoxyl sulfate and p-cresyl sulfate activate the intrarenal renin–angiotensin system, the TGF/Smad pathway, and possibly epithelial mesenchymal transformation. These effects are thought to induce inflammation and tissue injury and to accelerate the development of fibrosis (*Pan & Kang, 2018*). Uremic toxins have been shown to promote and further hasten kidney disease progression and cardiovascular disease (*Wu et al., 2011*; *Aronov et al., 2011*; *Lin et al., 2011*; *Sun, Chang & Wu, 2012*; *Mutsaers et al., 2015*; *Meijers & Evenepoel, 2011*). The microbiota in the large bowel can change after appendectomy and impaired growth of the microbiota may lead to dysbiosis (*Wu et al., 2015*). Dysbiosis caused by appendectomy may result in a pathogenic inflammatory effect and may increase the risk of CKD.

We performed subgroup analysis of the risk of CKD and ESRD in patients with appendectomy and matched participants (Tables 3 and 4). The interactions between sex, age group, comorbid conditions, and appendectomy for the risk of CKD and ESRD were not significant (all $P_{\text{interaction}} > 0.05$). However, the interaction effect of DM with appendectomy on ESRD or CKD had apparent inconsistencies. Appendectomy patients with concomitant DM had a significantly increased risk of ESRD (aHR 2.08, 95% CI [1.27–3.43], $P = 0.004$; Table 4) and the interactions between appendectomy and DM for ESRD risk were significant ($P = 0.022$). However, the risk of CKD and the interaction effect between appendectomy and DM for CKD risk was not obvious (Table 3). Patients with DM undergoing appendectomy have significantly more comorbidities and tend to have more postoperative complications (*Bach et al., 2016*). Inflammation is a cardinal pathogenic mechanism in diabetic nephropathy (*Barutta et al., 2015*; *Lim & Tesch, 2012*). We suggest that altered immune function and inflammation caused by appendectomy could worsen renal function in diabetic patients and increase the risk of ESRD. The high prevalence and low awareness of CKD in Taiwan (*Hsu et al., 2006*) could be a major factor to explain our finding that DM seems to be a risk factor for ESRD but not for CKD after appendectomy (Tables 3 and 4). In the cohort, we also demonstrated that appendectomy showed different effects across the stroke and statin-using subgroups, the interactions were insignificant (Fig. 3). In other words, there was no strong evidence of different effects of appendectomy in these subpopulations. Further studies are needed to confirm these findings.

## Limitations

This study had limitations. First, the NHIRD does not include information about smoking history, body mass index, family history of renal disease, blood pressure, lipid profile, glucose or uric acid concentrations, proteinuria, dietary habits, use of over-the-counter drugs or herbal remedies, or data on chronic glomerulonephritis or chronic interstitial nephritis. Although we performed propensity-score matching and adjusted for various confounders, these unmeasured confounders might have affected our results. Second, renal outcomes were mainly identified with ICD-9-CM codes. Because CKD severity and

estimated glomerular filtration rate were unknown, the number of patients with CKD and ESRD could be underestimated. However, ICD-9-CM codes are recognized as reliable indicators of CKD, ESRD, and comorbidities. Third, results derived from a retrospective cohort study are generally of lower statistical quality than those from prospective studies because of potential biases. Finally, the majority of Taiwan's population is of Chinese ethnicity; the results of this study may not be applicable to other populations.

## CONCLUSION

Understanding risk factors and implementing screening of at-risk populations will increase early detection of kidney disease, allowing initiation of treatment of modifiable risk factors for ESRD, along with appropriate treatment for CKD. To our knowledge, this study is the first to show that appendectomy is a significant risk factor for CKD and ESRD. Physicians should pay careful attention to renal function prognosis in appendectomy patients, especially in patients with DM. Clearly, this link needs to be validated in more specifically designed studies.

## ACKNOWLEDGEMENTS

We thank Rebecca Tollefson, DVM, from Edanz Group (http://www.edanzediting.com/ac) for editing a draft of this manuscript.

### Funding
This study was funded by grants 105-CCH-IRP-093 from the Changhua Christian Hospital Research Foundation. The funders had no role in study design, data collection and analysis, decision to publish, or preparation of the manuscript.

### Competing Interests
The authors declare there are no competing interests.

### Author Contributions
- Chin-Hua Chang, Chew-Teng Kor, Chia-Lin Wu, Ping-Fang Chiu, Jhao-Rong Li, Chun-Chieh Tsai, Teng-Hsiang Chang and Chia-Chu Chang conceived and designed the experiments, performed the experiments, analyzed the data, contributed reagents/materials/analysis tools, prepared figures and/or tables, authored or reviewed drafts of the paper, approved the final draft.

### Ethics
The following information was supplied relating to ethical approvals (i.e., approving body and any reference numbers):

This study was approved after a full ethical review by the Institutional Review Board (IRB) of Changhua Christian Hospital (approval number 151219). The IRB waived the need for consent. Data were accessed anonymously.

## Data Availability

The raw data are provided as a Supplemental File.

## Supplemental Information

Supplemental information for this article can be found online at http://dx.doi.org/10.7717/peerj.5019#supplemental-information.

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
