# Peer review of "Increased chronic kidney disease development and progression in diabetic patients after appendectomy: a population-based cohort study"

_PeerJ, doi:10.7717/peerj.5019_

## Round 0.1 · original submission · Major Revisions

Dera Dr. Chang

Your manuscript entitled " Increased chronic kidney disease development and progression in diabetic patients after appendectomy: A population-based cohort study" which you submitted to PeerJ, has been reviewed by the editor and 3 experts in the field.

I regret to inform you that the reviewers have raised some significant concerns that need to be addressed before the manuscript can be considered further. Since two reviewers do find some merit in the paper and findings are of potential clinical relevance, I would be willing to reconsider if you wish to undertake major revisions and resubmit.

If you decide to resubmit the revised version, please summarize all the improvements made in the new version and give answers to all critical points raised in the reviewers’ report in an accompanying letter. I strongly suggest to include a section that addresses the study's limitations. Bold statements should be avoided and the conclusion should be toned down considering alternative explanations. Finally, I and the reviewers had problems with opening the extension .sas7bdat and couldn’t check the raw data. Please, provide raw data in a more usual format (like excel).

Regards,

Stefano Menini

·

Basic reporting

The paper is written in clear English and is very understandable for readers. Introduction and background is sufficient, structure is appropriate. Raw data are available.

Experimental design

The research question has been answered. IRB reviewed. Impressive and stuffiest number of participants was included: 10383 patients and 41532 healthy people. Statistical analysis is appropriate.

Validity of the findings

This is an intrigue paper with new finding of link between appendectomy and development of CKD and ESRD, and particularly in patients with DM. DM is one of the leading causes of ESRD, and idea of close follow up after appendectomy for CKD development is clinically relevant.

Additional comments

The significant funding of interaction between appendectomy and DM for the risk of CKD/ESRD development are interesting and clinically relevant.

I was not sure about the duration of the study. In the abstract authors mentioned 2000-2012, but in the text: 2000-2013.
There are repetitions in Discussion section, which were already addressed in Results. There is no need for that. It should be corrected.

Table 1 needs a bigger font and not all abbreviations were spelled out (DM).

Reviewer 2 ·

Basic reporting

The Authors present a paper, in which they evaluate the association between appendectomy and increased risk to develop chronic Kidney disease (CDK) and end-stage renal disease (ESRD). The Authors conclude saying that appendectomy is a new risk factor for CKD because it contributes to intestinal dysbiosis in patients undergoing the surgery procedure. However, in my opinion, there are too many limitations that affect the paper results. First of all, considering that Taiwan has a high incidence and prevalence rates of CKD and ESRD, the missing information regard some important and predominant risk factors such as family history of renal disease, uric acid concentration, proteinuria, use of drugs and herbal medicine, and data on chronic glomerulonephritis and chronic interstitial nephritis are crucial to interpret the results correctly.

Experimental design

Data regarding well-established risk factors information for CKF and ESRD are missing.

Validity of the findings

In this study, the Authors hypothesize and conclude that appendectomy causes gut dysbiosis leading to CKD. However, no evidence is reported to demonstrate the correlation between the impairment of gut microbiota and appendectomy procedure. No uremic toxins are measured in the serum samples to demonstrate the link between appendectomy and gut dysbiosis.

The increased risk for CKD and ESRD in appendectomy patients are not supported by important information on risk factors.

How do The Authors demonstrate that the inflammation in diabetic patients is altered by appendectomy and not by a chronic dysmetabolic profile?

·

Basic reporting

Literature references, especially in the discussion section but also in the introduction, should be considerably expanded. Authors aim to demonstrate a potential link between appendectomy and the incidence of CKD, mediated by a putative appendectomy-induced gut microbiota dysbiosis. Although intriguing, the causative role of gut dysbiosis in the aetiology of CKD has not been demonstrated yet, and this research, although providing a novel and interesting result, does not demonstrate this mechanism. Authors should stress this aspect, dampening the speculations about this point. In CKD, a complex bi-directional relationship between gut microbiota and kidney is present, I suggest to expand the description of the scientific context, both in introduction and discussion sections (lines 76-84 and 275-282), providing more literature to support this hypothesis.

The manuscript is written in clear and correct English. Anyway, I would suggest to re-formulate long paragraphs (such that at lines 88-92) into shorter sentences in order to improve their readability.

Experimental design

Although the research topic is innovative and well conducted, the research gap should be better discussed and the research question must be completely revised. Indeed, this is an observational study whose aim cannot be to demonstrate that “appendectomy causes gut dysbiosis and that this dysbiosis results in CKD”. Authors should be aware that this is beyond the possibilities of their study and should better discuss this point in the study limitations. It is necessary to modify the sentence at lines 98-99. I would rather state that authors aim to evaluate the presence of an association between an increased CKD risk in those patients undergoing a past appendectomy.
I recommend authors to contextualise the putative link with gut microbiota dysbiosis as their own speculation, also because data on gut microbiota composition or at least on uremic toxins are not included in their research. For this reason, the presented results do not allow to draw conclusions on microbial dysbiosis as the causative link between the two pathological events. Any link with gut microbiota dysbiosis should be moved to the discussion and referred to only at speculative level.

Please provide a justification for the use of a 1:4 ratio instead of a one-to-one matching between the appendectomy and the propensity score-matched cohort. This is not clearly explained.

Validity of the findings

The present research demonstrates an association between a past appendectomy and an incident CKD, especially in diabetic patients.
In spite of the novelty of the research, there are some apparently inconsistent results which are worthy of authors’ comments in the discussion:

1. along with the increased CKD and ESRD occurrence, appendectomy patients seem to display a reduced mortality (Table 1)
2. the risk for CKD results higher only in male patients and, strangely, only in those without comorbidities and not with comorbid conditions (Table 3)
3. diabetes seems to be a risk factor only for ESRD and not for CKD (I would expect a similar risk also for CKD) (Table 4)
4. although not significant, the interaction analysis seem to suggest that comorbid factors such as stroke and use of statins abolish the association between appendectomy and increased CKD or ESRD risk (Fig 3), resulting as a kind of “protective” factors.

I suggest to provide, along with the “microbiota hypothesis”, other potential hypotheses explaining the association they found. For example, the authors could include in their discussion other possible links between appendectomy and incident CKD, such as the presence of a previous renal subclinical alteration (Tal 2017, doi: 10.1002/jum.14480) or the probability of AKI after appendectomy (Kim 2014, doi: 10.1213/ANE.0000000000000425) or the evidence that diabetic patients basally present a complicated clinical frame (Bach 2016, PMID: 27657594), even if a reference to the latter point is present in the manuscript (290-294). I recommend to further expand this point. Furthermore, dietary habits, in particular a scarce intake of dietary fiber, could represent a common risk factor for both conditions. Please add all these points to the discussion and to the limitations of the study (dietary habits).

Please correct CKD with ESRD at line 234.

---

## Round 0.2 · Minor Revisions

Dear Dr. Chang,

Thank you for your resubmission. I have now received reports from all three reviewers who are generally supportive of publication. However, reviewer 3 suggested minor modifications. Accordingly, I invite you to address the remaining reviewer' s comments and recommendations.

With Kind regards,

Stefano Menini

·

Basic reporting

no comment

Experimental design

no comment

Validity of the findings

no comment

Additional comments

All 3 previous comments have been addressed.

Reviewer 2 ·

Basic reporting

I have no further comments, and all the issues raised in the first revision process have been successfully addressed. The manuscript is improved.

Experimental design

I have no further comments, and all the issues raised in the first revision process have been successfully addressed.

Validity of the findings

I am satisfied with the modifications made by the authors. They provide a novel and interesting resultand although remain research gaps and limitations. I have no further critical comments.

Additional comments

The manuscript is improved, I appreciate the authors' response to my comments.

·

Basic reporting

The manuscript is written in clear and correct English, and the revisions made are ok. Please change “diabetics” in “diabetes” (line 304). Raw data are available.

Experimental design

The authors solved many of the raised questions.
I would rather rephrase the sentence in lines 96-98 into “we evaluated the correlation between a past appendectomy and the occurrence of CKD and ESRD”.

Validity of the findings

The authors solved many of the raised questions.
Just few remaining issues:

Table 1 - along with the increased CKD and ESRD occurrence, appendectomy patients seem to display a reduced mortality.
Thank you for the further analysis you performed, that I suggest to include in the manuscript.

Table 4 - Diabetes seems to be a risk factor only for ESRD and not for CKD (I would expect a similar risk also for CKD).
Ok, but I suggest to add a comment in the manuscript about the apparent inconsistence of the interaction between diabetes and ESRD and not with CKD.

The revisions made in the discussion section are ok. I suggest to re-add a reference already present in the first version of the manuscript (Sabatino 2017, doi: 10.1007/s11892-017-0841-z), thoroughly discussing gut dysbiosis and inflammation in CKD and diabetes, as underlying and linking factors between the two pathologies. In the context of this manuscript, it could be useful to the authors to support a putative hypothesis explaining the association they found between appendectomy, ESRD and diabetes. Indeed, appendectomy could have a cumulative effect on the pre-existing dysbiosis in diabetic patients, and increase the risk of ESRD occurrence. Clearly, this link needs to be validated in more specifically designed studies.

---

## Round 0.3 · Minor Revisions

Dear Dr. Chang,

Thank you for your resubmission. Reviewer 3 suggested minor modifications. Accordingly, I invite you to address the reviewer' s comments and recommendations.

The manuscript needs language editing. Awkward phrasing like that reported by reviewer 3 should be fixed to ensure that your international audience can clearly understand your text. I suggest that you have a native English speaking colleague review your manuscript before resubmission.

Please, pay more attention to the reviewer’s 3 criticisms, which I share, as this invitation to resubmit is not accompanied by a commitment to publish

Regards,

Stefano Menini

·

Basic reporting

Authors: We thank the reviewer for their kind reminders. We revised “diabetics” to “diabetes” in line 304.

The correction was made in a wrong place, please correct “diabetics” in “diabetes” at line 312 (I refer to the Word manuscript with tracked changes) and leave the adjective “diabetic” referred to "patients" at line 317 (the original version was correct).
I suggest to perform a last language revision by a native English speaker, especially for the added revisions (for example, the sentence “The interaction effect of DM with appendectomy was appeared inconsistence, the observable subgroup effect on ESRD, but not on CKD” is not understandable).

Experimental design

No comment

Validity of the findings

The revisions of Table 1 and conclusion are ok.
Please correct the syntax of the sentence "The interaction effect of DM with appendectomy was appeared inconsistence, the observable subgroup effect on ESRD, but not on CKD."

Thank you for adding the suggested reference. Anyway, in Sabatino et al, a causative relationship between appendectomy and dysbiosis is not reported.

For this reason, a modification of the sentence:

"The microbiota in the large bowel can change after appendectomy and lead to dysbiosis.9 Evidence suggests that appendectomy impairs gut microbiota and results in a pathogenic inflammatory effect.18”

into:

“The microbiota in the large bowel can change after appendectomy and lead to dysbiosis9. Gut dysbiosis and inflammation are underlying and linking factors between CKD and diabetes18. Appendectomy could have a cumulative effect on the pre-existing dysbiosis in diabetic patients, increasing the risk of ESRD occurrence.”

would be more appropriate.

---

## Round 0.4 · accepted · Accept

Dear Dr. Chang,

Thank you for submitting a revised version of your manuscript. I am pleased to inform you that your manuscript is accepted for publication in PeerJ in its current form and will now be forwarded to the production editor for publication.

I thank all reviewers for their effort in improving the manuscript.

Yours sincerely,

Stefano Menini

·

Basic reporting

The authors solved all the raised questions

Experimental design

No Comment

Validity of the findings

The authors solved all the raised questions